# Biomechanical evaluation of high tibial osteotomy plate with internal support block using finite element analysis

**Jesse Chieh-Szu Yang**[1,2], **Kuan-Yu Lin**[3,4], **Hsi-Hsien Lin**[1,5], **Oscar K. Lee**[2,6,7]*

1 Department of Orthopedics and Traumatology, Taipei Veterans General Hospital, Taipei, Taiwan,
2 Institute of Clinical Medicine, National Yang-Ming University, Taipei, Taiwan, 3 Department of Orthopedics, Kaohsiung Veterans General Hospital, Kaohsiung City, Taiwan, 4 Department of Nursing, Meiho University, Neipu, Pingtung, Taiwan, 5 School of Medicine, National Yang-Ming University, Taipei Taiwan, 6 Department of Orthopedics, China Medical University Hospital, Taichung, Taiwan, 7 Department of Orthopaedics and Traumatology, and Institute for Tissue Engineering and Regenerative Medicine, The Chinese University of Hong Kong, Hong Kong, China

* cyl710521@gmail.com

**Data Availability Statement:** All relevant data are within the manuscript.

**Funding:** The authors received no specific funding for this work.

## Abstract

### Background/Objective

High tibial osteotomy (HTO) is a common treatment for medial knee arthrosis. However, a high rate of complications associated with a plate and a significant loss of correction have been reported. Therefore, an internal support block (ISB) is designed to enhance the initial stability of the fixation device that is important for successful bone healing and maintenance of the correction angle of the osteotomy site. The purpose of this study was performed to examine if an internal support block combined with a plate reduces the stress on the plate and screw area.

### Methods

Finite element models were reconstructed following three different implant combinations. Two loading conditions were applied to simulate standing and initial sit-to-stand postures. Data analysis was conducted to evaluate the axial displacement of the posteromedial tibial plateau, which represents the loss of the posteromedial tibial plateau in clinical observation. Moreover, the stresses on the bone plate and locking screws were evaluated.

### Results

Compared to the TomoFix plate, the ISB reduced the axial displacement by 73% and 76% in standing and initial sit-to-stand loading conditions, respectively. The plate with an ISB reduced stress by 90% on the bone plate and by 73% on the locking screw during standing compared to the standalone TomoFix plate. During the initial sit-to-stand loading condition, the ISB reduced the stress by 93% and 77% on the bone plate and the locking screw, respectively.

**Competing interests:** The authors have declared that no competing interests exist.

## Conclusion

The addition of the PEEK block showed a benefit for structural stability in the osteotomy site. However, further clinical trials are necessary to evaluate the clinical benefit of reduced implant stress and the internal support block on the healing of the medial bone tissue.

## Introduction

Open-wedge high tibial osteotomy (OWHTO) is a surgical treatment for symptomatic malalignment in young and active patients [1, 2]. The initial stability of the fixation plate is critical for the successful maintenance of correction and the fusion of the osteotomy site. There are many different implants used for correcting varus deformity of the knee associated with medial compartment osteoarthritis [3–6]. However, a high rate of plate-related complications and a significant loss of correction have been reported [7–10]. Several biomechanical studies were performed to investigate the mechanical stability by finite element (FE) analyses [11–14].

The goal of the current study was to compare the biomechanical features of 3 different fixation techniques in medial open-wedge HTO. The approaches included Medial High Tibial Plate (MHTP) fixation, MHTP with cannulated lag screw (CLS) fixation, and MHTP with internal support block (ISB) fixation.

## Materials and methods

### Generation of osteotomy finite element model

The intact left tibial model was obtained by the scan of the fourth generation Sawbones #3401 (SKU 3973, Pacific Research Laboratories, Inc., Vashon, WA, USA). The contours of the cortical and cancellous bone were used to generate the solid model in the SolidWorks CAD software (Solid Works Corp., Boston, U.S.A.). The tibial shaft is perpendicular to the ground in the sagittal plane, with a 3° varus tilting according to the general concept of the lower limb anatomy in the standing posture. Biplanar tibial tubercle preserving osteotomy was simulated for approximately 10 mm expansion at the medial osteotomy site (Fig 1) as the designated correction in OWHTO.

### Generation of osteotomy finite element model with three different implant combination

This study employed an automatic mesh generation algorithm with Simulation Version 2010 software (SolidWorks Corporation, Concord, MA, USA). The following three different implant combinations were inserted into the osteotomy model (Fig 2): (A) the conventional fixator, the TomoFix (Depuy Synthes, PA, USA) was placed at the anteroposterior region of the osteotomy model, denoted as "MHTP". A total of 8 locking screws (diameter, 5 mm) were fixed in all the locking holes on the plate; (B) a cannulated lag screw (Diameter of 8mm; Stryker, MI, USA) was applied in the MHTP model to simulate the technique of opposite screw insertion, denoted as "CLS". The opposite screw was inserted from the lateral cortex to the region beneath the medial-lateral plateau, in the orientation of approximately 50 degrees oblique in the coronal plane and 38.5 degrees oblique in the transverse plane; (C) a PEEK internal block was applied in the MHTP model to determine how an internal block contributes to the stability of osteotomy model, denoted as "ISB".

To simplify the finite element analysis, the lower half of the surrogate bones and the threads on the screws (including the locking screws and the cannulated lag screw) were all removed.

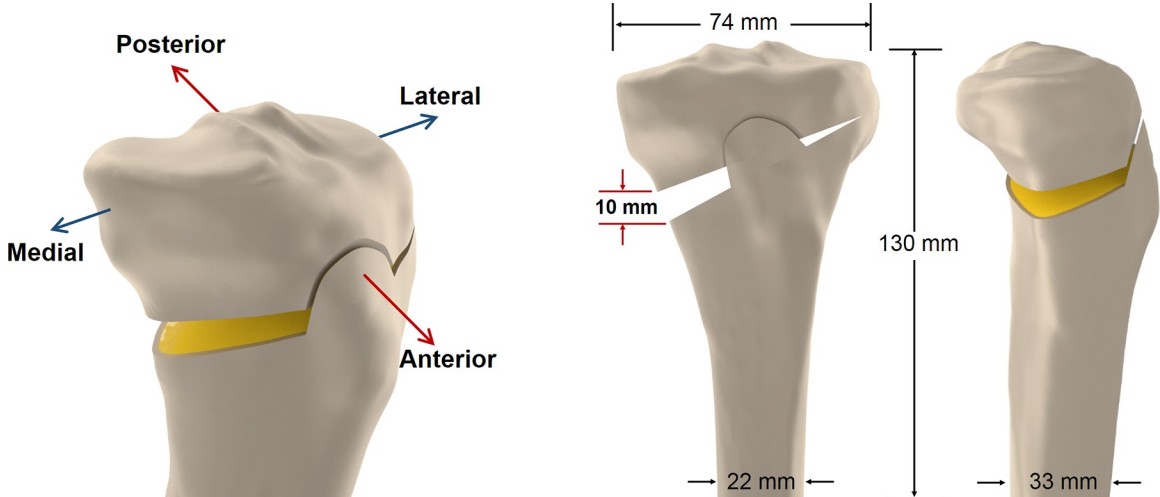

**Fig 1. Orientation definition.** (a) Orientation definition; (b) The coronal view of the solid model representing a biplanar high tibial osteotomy; (c) The sagittal view of the solid model representing a biplanar high tibial osteotomy.

The isotropic linear homogeneous elastic material properties were assigned to different parts of the model according to the previous literature [15] as shown in Table 1. A Young's modulus of 17,000 and 300 MPa, and Poisson's ratio of 0.36 and 0.3 were defined for the cortical bone and the cancellous bone, respectively. The plate, locking screw, and cannulated lag screw were all defined as titanium alloy with homogeneous and linear elastic properties, and the values of Young's modulus and Poisson's ratio were 113 GPa and 0.33, respectively. The internal block was defined as PEEK with Young's modulus of 3.5 GPa and Poisson's ratio of 0.3 [16].

## Loading and boundary conditions

Two different loading conditions were applied in all three models to simulate the loads in the standing posture and at the initial stage of the sit-to-stand movement. For the standing posture, a 600 N axial compressive load was applied on the full tibial plateau and the loading ratio of the medial and lateral plateau was 60 and 40%, respectively [15] (Fig 3). An additional

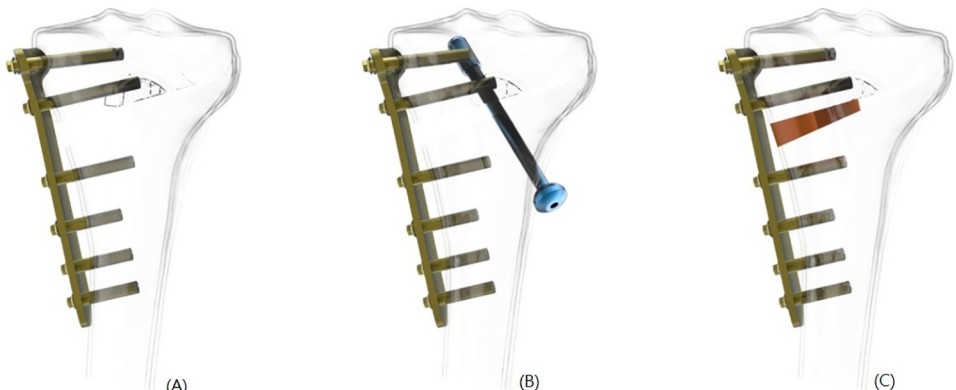

**Fig 2. The osteotomy model.** The osteotomy model with three different implant combinations: (a) The TomoFix (MHTP); (b) The TomoFix with an 8 mm cannulated lag screw (CLS); (c) the TomoFix with a PEEK internal support block (ISB).

**Table 1. Material properties specified in the finite element models.**

| Material | Elastic modulus (MPa) | Poisson's ratio |
|---|---|---|
| **Cancellous bone** | 17000 | 0.36 |
| **Cortical bone** | 300 | 0.30 |
| **Titanium Alloy** | 113000 | 0.33 |
| **PEEK** | 3.5 | 0.30 |

600-N axial compressive load was applied on the posterior half of the tibial plateau only to simulate the load at the initial stage of the sit-to-stand movement [17]. For the osteotomy model with the cannulated lag screw, a pretension force of 100 N was applied on the 8.0-mm lag screws. The screw-plate and block-bone interface was assumed to be bonded without separating and sliding. Fully constrained was applied to the distal end of the tibial osteotomy model (Fig 3).

### Data analysis

Data analysis was conducted to evaluate the effect of the opposite screws and internal block on the stabilization. The axial displacement of the posteromedial tibial plateau was analyzed and compared with the osteotomy model. Each maximum Von Mises stress at the plate and locking screws was calculated in two loading conditions for three different models.

## Results

### Differences in axial displacement posteromedial tibial plateau

Relative loss of posterior reduction after loading was found at the posteromedial region of the tibial plateau. The values of the axial displacement were normalized in the MHTP model (with Tomofix) for the two different loading conditions. Compared with the values of the MHTP model in standing and sit-to-stand loads, the axial displacement of the tibial plateau after the lag cannulated screw insertion decreased by 11% and 18%, respectively (Fig 4). The axial displacement of the tibial plateau after the internal block insertion decreased by 73% and 76%, respectively (Fig 4).

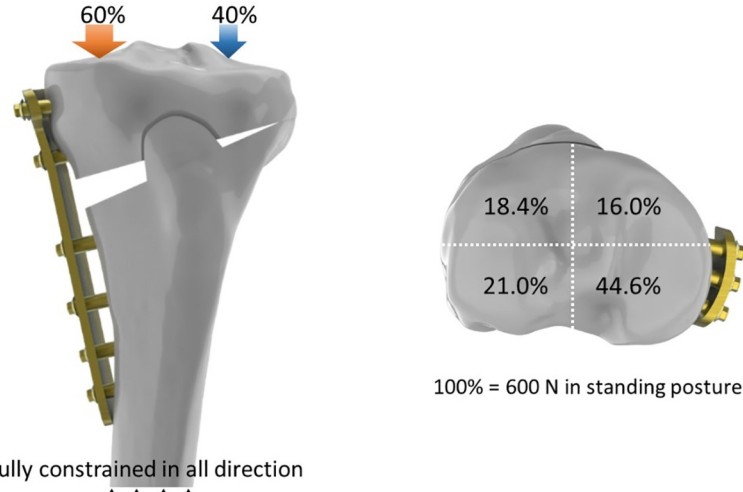

**Fig 3. Illustrations showing the ratio of the axial loads.** Illustrations showing the ratio of the axial loads on the quadrants of the proximal tibia.

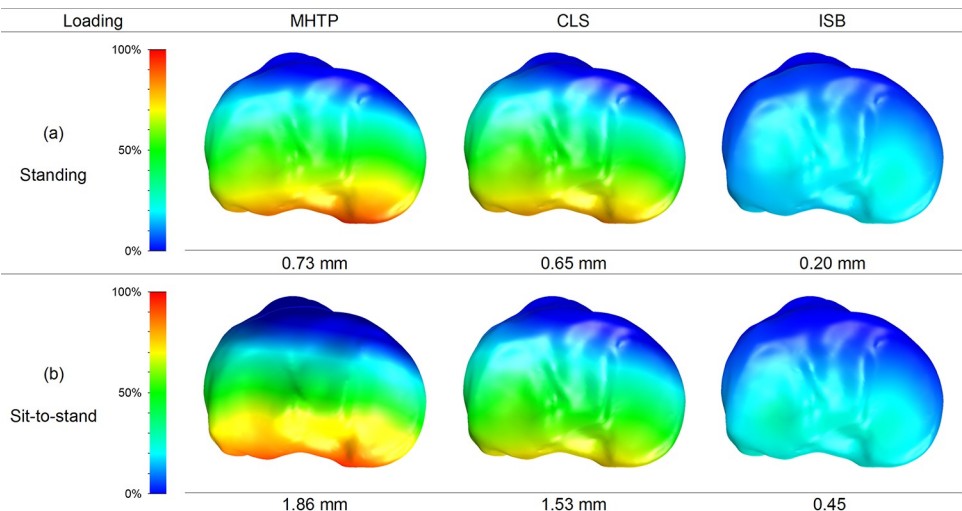

**Fig 4. The percentages of loss of posteromedial reduction.** The percentages of loss of posteromedial reduction on the tibial plateau, normalized by the magnitude in the osteotomy model.

## Stresses on the bone plate

The percentile differences of the maximum von Mises stress on the bone plate were calculated as (σCLS-σMHTP)/σMHTP or (σISB-σMHTP)/σMHTP in two different loading conditions. The maximum stress of the bone plate after the lag cannulated screw insertion decreased by 20% and 26% in standing and sit-to-stand, respectively (Fig 5A). The maximum stress of the bone plate after the internal block insertion decreased by 90% and 93% in standing and sit-to-stand, respectively (Fig 5B).

## Stress on locking screws

The percentile differences of the maximum von Mises stress on the locking screws were calculated as (σCLS-σMHTP)/σMHTP or (σISB-σMHTP)/σMHTP in two different loading conditions. The maximum stress of the locking screws after the lag cannulated screw insertion

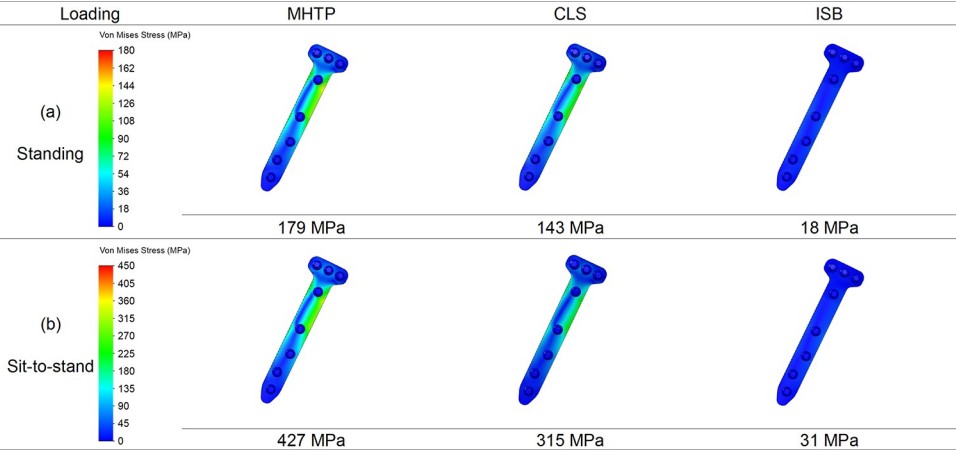

**Fig 5. The stress patterns and values on the bone plate.** The stress patterns and values on the bone plate in three models in the standing and sit-to-stand loading conditions.

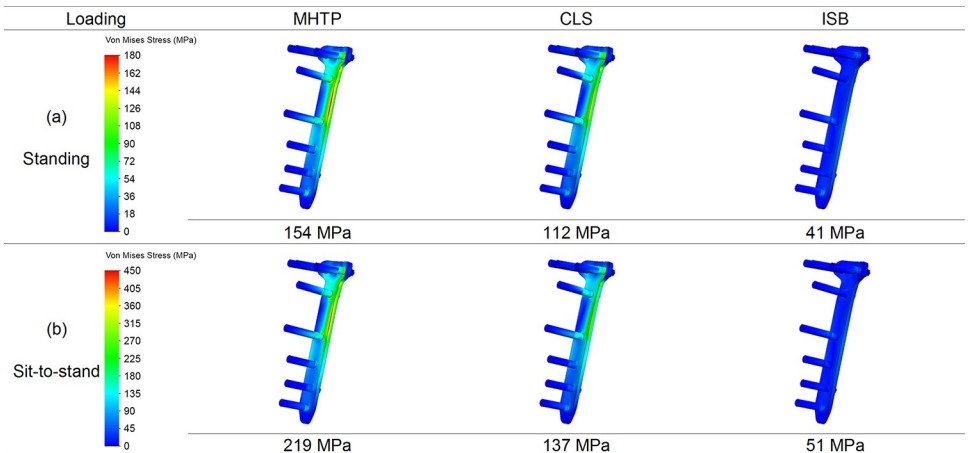

**Fig 6. The stress patterns and values on the locking screws.** The stress patterns and values on the locking screws in three models in the standing and sit-to-stand loading conditions.

decreased by 27% and 37% in standing and sit-to-stand, respectively (Fig 6A). The maximum stress of the locking screws after the internal block insertion decreased by 73% and 77% in standing and sit-to-stand, respectively (Fig 6B).

## Discussion

The knee joint is one of the largest and most heavily loaded joints of the human body during daily activities [18]. The fixation device of HTO surgery is used to stabilize the opening and enhance bone union. This study used finite element analysis to evaluate the effects of the construct stress of the locking screw, plate, and wedge micromotion. For the ISB model, the screw, plate, and bone stresses of the TomoFix plate were respectively 73%, 90%, and 79% less than those of the MHTP model (Fig 7).

Previous FE and biomechanical studies on the medial fixation device demonstrated good stability [19, 20]. The purpose of this study was to compare a TomoFix plate, a TomoFix plate with a lag cannulated screw, as well as a PEEK internal block inserted to support the medial opening gap, by using computational simulations. Our results showed that the stress on the implants in the CLS was lower than that in the MHTP. This observation was also reported in

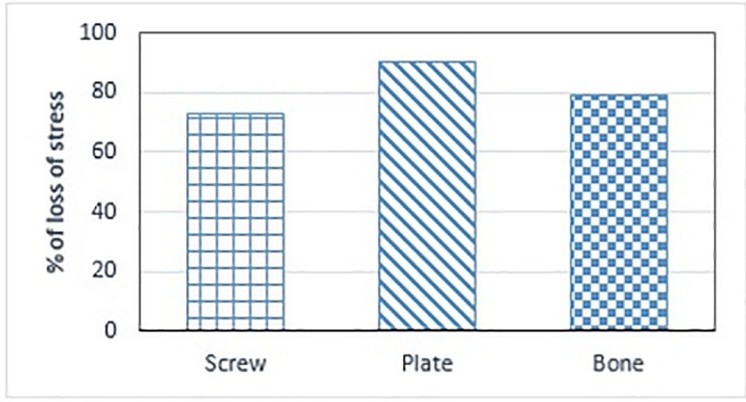

**Fig 7. Percentages of loss of stress.** Percentages of loss of stress on the screw, plate, and bone, respectively.

an FE analysis in a previous study [21]. The study demonstrated that a TomoFix with a lag cannulated screw provided a better anchorage than the plate alone, thereby displacing the stresses transmitted by body weight. The PEEK block reduced the downward displacement of the proximal tibia and screw. Therefore, the plate stress is reduced to prevent implant broken and to enhance the stability of the tibia after high tibial osteotomy.

Several limitations of the study should be further considered. Firstly, the material properties of all components in the current FE model were assigned as homogenous, isotropic, and linear elastic. This study focused on the effect of different fixation methods of HTO. The influence of simplified material properties of bony structure on simulated results can be comparatively minor. In addition, only a single Sawbone model has been considered in this study. The different geometries of real tibial bone structures would possibly be influential to biomechanical behavior after HTO. Secondly, the findings could be different in a real clinical situation even though simulated conditions reflecting clinical and surgical situations were assessed in all cases. Especially, it is difficult to define the contact conditions of a real situation by using the friction coefficient of general contact conditions. Moreover, only the standing and sit to stand loading conditions were considered. The effect of fixation approaches in walking was not considered. Lastly, The theoretical advantage in our simulated study is difficult to predict directly whether surgical complications can be avoided in the actual situation.

## Conclusion

Compared to conventional fixation instrumentation, the plate/screw stress was redistributed and reduced with the combination of a PEEK internal block. However, further investigation on in vitro biomechanical tests and clinical trials will be necessary to determine whether this internal PEEK block can reduce the fracture risk on the implant and enhance the bone fusion rate in the osteotomy site.

## Author Contributions

**Conceptualization:** Jesse Chieh-Szu Yang, Kuan-Yu Lin, Hsi-Hsien Lin, Oscar K. Lee.

**Data curation:** Jesse Chieh-Szu Yang, Kuan-Yu Lin, Hsi-Hsien Lin, Oscar K. Lee.

**Formal analysis:** Jesse Chieh-Szu Yang, Kuan-Yu Lin, Hsi-Hsien Lin, Oscar K. Lee.

**Investigation:** Jesse Chieh-Szu Yang, Kuan-Yu Lin, Hsi-Hsien Lin, Oscar K. Lee.

**Methodology:** Jesse Chieh-Szu Yang, Kuan-Yu Lin, Hsi-Hsien Lin, Oscar K. Lee.

**Project administration:** Jesse Chieh-Szu Yang, Kuan-Yu Lin, Hsi-Hsien Lin, Oscar K. Lee.

**Resources:** Jesse Chieh-Szu Yang, Kuan-Yu Lin, Hsi-Hsien Lin, Oscar K. Lee.

**Software:** Jesse Chieh-Szu Yang, Kuan-Yu Lin, Hsi-Hsien Lin, Oscar K. Lee.

**Supervision:** Jesse Chieh-Szu Yang, Kuan-Yu Lin, Hsi-Hsien Lin, Oscar K. Lee.

**Validation:** Jesse Chieh-Szu Yang, Kuan-Yu Lin, Hsi-Hsien Lin, Oscar K. Lee.

**Visualization:** Jesse Chieh-Szu Yang, Kuan-Yu Lin, Hsi-Hsien Lin, Oscar K. Lee.

**Writing – original draft:** Jesse Chieh-Szu Yang, Kuan-Yu Lin, Hsi-Hsien Lin, Oscar K. Lee.

**Writing – review & editing:** Jesse Chieh-Szu Yang, Kuan-Yu Lin, Hsi-Hsien Lin, Oscar K. Lee.

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
