## [Decision Letter · Decision Letter 0]

23 Dec 2020

PONE-D-20-23602

Biomechanical Evaluation of High Tibial Osteotomy Plate with Internal Support Block Using Finite Element Analysis

PLOS ONE

Dear Dr. Lee,

Thank you for submitting your manuscript to PLOS ONE. After careful consideration, we feel that it has merit but does not fully meet PLOS ONE’s publication criteria as it currently stands. Therefore, we invite you to submit a revised version of the manuscript that addresses the points raised during the review process.

The reviewers comments will help to improve the manuscript. Please, address any point.

We look forward to receiving your revised manuscript.

Kind regards,

Hans-Peter Simmen, M.D., Professor of Surgery

Academic Editor

PLOS ONE

Journal Requirements:

3. We noticed you have some minor occurrence of overlapping text with the following previous publication, which needs to be addressed:

- https://link.springer.com/article/10.1007/s00402-018-2918-9 ("Role of an anatomically contoured plate and metal block for balanced stability between the implant and lateral hinge in open-wedge high-tibial osteotomy", Young Woong Jang, DoHyung Lim, Hansol Seo, Myung Chul Lee, O-Sung Lee & Yong Seuk Lee)

In your revision ensure you cite all your sources (including your own works), and quote or rephrase any duplicated text outside the methods section. Further consideration is dependent on these concerns being addressed.

Reviewers' comments:

Reviewer's Responses to Questions

**Comments to the Author**

1. Is the manuscript technically sound, and do the data support the conclusions?

Reviewer #1: Yes

Reviewer #2: Partly

2. Has the statistical analysis been performed appropriately and rigorously? 

Reviewer #1: N/A

Reviewer #2: I Don't Know

3. Have the authors made all data underlying the findings in their manuscript fully available?

Reviewer #1: Yes

Reviewer #2: No

4. Is the manuscript presented in an intelligible fashion and written in standard English?

Reviewer #1: Yes

Reviewer #2: Yes

5. Review Comments to the Author

Reviewer #1: The manuscript evaluates the effects of an internal support block on the stresses on bone plate and locking screws. This manuscript covers an interesting topic and is worth considering. However, there are some important issues about the details of the research. Critiques of the revised manuscript and suggestions for the authors are given below:

1. (Lane 69) The geometric information of the left tibial sawbone is not clear. Is this model developed based on medical images? If so, basic information about the patient/volunteer is needed, such as, age, weight, injury history, etc. If this model is previously published, please add a reference, or clarify what is #3401; Sawbones, WA, United States.

2. (Lane 96) Please justify that isotropic linear homogeneous elastic material model is appropriate. It might be more accurate to model the bone as elastic-plastic material according to other studies.

For example:

Tippanagoudar, Naveen, and A. Krishna. "Finite element analysis of tibia bone." Int J Eng Sci Comput 8.12 (2018): 19534-7.

Untaroiu, Costin D., Neng Yue, and Jaeho Shin. "A finite element model of the lower limb for simulating automotive impacts." Annals of biomedical engineering 41.3 (2013): 513-526.

3. (Lane 111) Please provide a reference for the initial load of sit-to-stand movement (600N axial load).

4. (Lane 120) It is unclear why the displacement of the posteromedial tibial plateau, stresses of the bone plate and locking stews were selected to be compared in this study. What are the effects of these parameters on clinical treatments?

5. (Lane 126-149) The results were only compared relatively (only percentile differences were shown). Please also show the absolute values. For example, the maximum stress of the bone plate. This data will provide more confidence to this study.

6. (Figure 1) It will be also helpful to show the size of the bone in figure 1, not only the size of the opening wedge.

Reviewer #2: In this evaluation the authors have compared three different types of fixation after high tibial osteotomy (HTO) by a finite element model.

The evaluation is purely based on a computed simulation. In their results the authors demonstrate that with an internal support block that is placed into the osteotomy the axial displacement is reduced by 18% in standing and 100% in a sit-to-stand loading condition.

Critical comment

It is difficult to say whether finite element models are able to simulate complex biomechanical conditions like weight bearing and loading after HTO.

The authors have compared three different models of high tibial osteotomy. Since the conclusions of the study are based purely on the simulation of loading the HTO, the authors have to demonstrate more in detail the reliability of their calculations.

The first model is an open osteotomy without any additional implants but plate and screws and is comparable to date standard procedure.

The second construct is an osteotomy with an additional lag screw that is placed with a 90° angulation to the osteotomy. The role of the lag screw is difficult to understand. This lag screw augments the compression forces on the osteotomy. As a lag screw this screw does not give any additional stability to the construct.

A) Therefore the authors have to explain why they have analysed this construct.

In their third model they have virtually placed an internal block of PEEK in the osteotomy to augment the stability with respect to compression forces. For the mechanical stability it makes a difference if the block is placed exactly on the cortex of the medial tibia or in the cancellous bone area.

B) It should be explained where the block is placed exactly in the tibial area.

A last question is whether a simple compression by standing or by a sit-to-stand procedure is simulated or if repeated loadings have been calculated. In order to simulate the real stress on such a construct repeated loading has to be taken into consideration. Only by such a biomechanical simulation it can be excluded that the internal block does not lose function when it is continuously pressed into the bone.

C) Is the model calculated with continuous loading and unloading?

6. PLOS authors have the option to publish the peer review history of their article (what does this mean?). If published, this will include your full peer review and any attached files.

Reviewer #1: No

Reviewer #2: No

---

## [Author Response · Author response to Decision Letter 0]

4 Feb 2021

Replies to reviewer #1

We would like to thank the reviewer for the great assistance in revising our manuscript entitled “Biomechanical evaluation of high tibial osteotomy plate with internal support block using finite element analysis” (PONE-D-20-23602). The comments were very helpful for making this manuscript more presentable. We have reviewed the comments carefully and have made corrections accordingly. All the newly added texts are underlined and marked in blue color in the revised manuscript. We hope the reviewer will find the revision satisfactory. Below, we reply to the reviewer's comments point-by-point.

1. (Lane 69) The geometric information of the left tibial sawbone is not clear. Is this model developed based on medical images? If so, basic information about the patient/volunteer is needed, such as, age, weight, injury history, etc. If this model is previously published, please add a reference, or clarify what is #3401; Sawbones, WA, United States.

Reply: Thank you for your comment. Sawbones #3401 (Pacific Research Laboratories, Inc., Vashon, WA, USA) is the fourth generation composite tibia designed to mimic the human tibia and usually used in the biomechanical test. The 3D CAD model was created from a scanned composite tibia obtained by Pacific Research Laboratories. A more detailed description of the 3D model of Sawbones #3401 has been added in the revised manuscript (Lane 69).

2. (Lane 96) Please justify that isotropic linear homogeneous elastic material model is appropriate. It might be more accurate to model the bone as elastic-plastic material according to other studies. For example: Tippanagoudar, Naveen, and A. Krishna. “Finite element analysis of tibia bone.” Int J Eng Sci Comput 8.12 (2018): 19534-7. Untaroiu, Costin D., Neng Yue, and Jaeho Shin. "A finite element model of the lower limb for simulating automotive impacts." Annals of biomedical engineering 41.3 (2013): 513-526.

Reply: Thank you for your comment. We agree with the reviewer that it might be more accurate to model the bone as elastic-plastic material. Therefore, the additional limitations described above were added in the revised manuscript (Lane 196).

3. (Lane 111) Please provide a reference for the initial load of sit-to-stand movement (600N axial load).

Reply: Thank you for your comment. The referenced literature for the loading condition of the sit-to-stand movement was added in the revised manuscript (Lane 267, Reference #17).

4. (Lane 120) It is unclear why the displacement of the posteromedial tibial plateau, stresses of the bone plate and locking screws were selected to be compared in this study. What are the effects of these parameters on clinical treatments?

Reply: The change in height at the posteromedial tibial plateau was the index of construct stability. Maximal von Mises stresses at screws and plates were the indices of implant and bone failure. The above three indices were chosen because stable correction, early full weight-bearing, and bone union after medial open-wedge HTO are essential to achieve the goal of enhanced recovery after surgery.

5. (Lane 126-149) The results were only compared relatively (only percentile differences were shown). Please also show the absolute values. For example, the maximum stress of the bone plate. This data will provide more confidence to this study.

Reply: Thank you for your comment. The absolute values have been added to Figure 4 to Figure 6. Additionally, some numbers of the percentile differences of results have been corrected in the revised manuscript.

6. (Figure 1) It will be also helpful to show the size of the bone in figure 1, not only the size of the opening wedge. 

Reply: Thank you for your comment. More information about the tibia size has been added to Figure 1.

Replies to reviewer #2

We would like to thank the reviewer for the great assistance in revising our manuscript entitled “Biomechanical evaluation of high tibial osteotomy plate with internal support block using finite element analysis” (PONE-D-20-23602). The comments were very helpful for making this manuscript more presentable. We have reviewed the comments carefully and have made corrections accordingly. All the newly added texts are underlined and marked in blue color in the revised manuscript. We hope the reviewer will find the revision satisfactory. Below, we reply to the reviewer's comments point-by-point.

1. It is difficult to say whether finite element models are able to simulate complex biomechanical conditions like weight bearing and loading after HTO.

Reply: Thank you for your comment. The additional limitation was added in the revised manuscript (Lane 202).

2. The authors have compared three different models of high tibial osteotomy. Since the conclusions of the study are based purely on the simulation of loading the HTO, the authors have to demonstrate more in detail the reliability of their calculations.

Reply: Thank you for your comment. The following figures show the validation of the current study in terms of result convergence and construct stiffness. The construct stiffness of MHTP converges to 1644 N/ mm until the element number reached about 262, 582 (left of the figure below). For the convergent stiffness, the numerical was within the range of the previous experimental results (right of the figure below). This indicates that good agreement is achieved, and the finite-element model is validated for further analyses.

3. The first model is an open osteotomy without any additional implants but plate and screws and is comparable to date standard procedure. The second construct is an osteotomy with an additional lag screw that is placed with a 90° angulation to the osteotomy. The role of the lag screw is difficult to understand. This lag screw augments the compression forces on the osteotomy. As a lag screw this screw does not give any additional stability to the construct. (A) Therefore the authors have to explain why they have analysed this construct.

Reply: In the previous FEA [Yang et al., JOT, 2020] and sawbones experimental studies [Yang et al., PLOS ONE, 2020], Yang et al. reported that structural stability and durability can be improved if followed by supplemental screw insertion in medial open wedge HTO. Therefore, the present study evaluated the potential advantage of block insertion compared to the opposite screw insertion in medial open-wedge HTO.

4. In their third model they have virtually placed an internal block of PEEK in the osteotomy to augment the stability with respect to compression forces. For the mechanical stability it makes a difference if the block is placed exactly on the cortex of the medial tibia or in the cancellous bone area. (B) It should be explained where the block is placed exactly in the tibial area.

Reply: The internal spacer was designed with a banana-shaped and placed posterior medial region of the medial open wedge side as shown in the following figure.

5. A last question is whether a simple compression by standing or by a sit-to-stand procedure is simulated or if repeated loadings have been calculated. In order to simulate the real stress on such a construct repeated loading has to be taken into consideration. Only by such a biomechanical simulation it can be excluded that the internal block does not lose function when it is continuously pressed into the bone. (C) Is the model calculated with continuous loading and unloading?

Reply: Only the standing and sit to stand loading conditions were considered in this study. Therefore, continuous loading and unloading conditions were not applied in our FE analysis. Based on the observation in the current study, further in vitro biomechanical tests and clinical trials are planned for clariﬁcation and veriﬁcation in reality.

---

## [Editor Report · Decision Letter 1]

8 Feb 2021

Biomechanical Evaluation of High Tibial Osteotomy Plate with Internal Support Block Using Finite Element Analysis

PONE-D-20-23602R1

Dear Dr. Lee,

We’re pleased to inform you that your manuscript has been judged scientifically suitable for publication and will be formally accepted for publication once it meets all outstanding technical requirements.

Kind regards,

Hans-Peter Simmen, M.D., Professor of Surgery

Academic Editor

PLOS ONE
---

## [Editor Report · Acceptance letter]

15 Feb 2021

PONE-D-20-23602R1 

Biomechanical Evaluation of High Tibial Osteotomy Plate with Internal Support Block Using Finite Element Analysis 

Dear Dr. Lee:

I'm pleased to inform you that your manuscript has been deemed suitable for publication in PLOS ONE. Congratulations! Your manuscript is now with our production department. 

Kind regards, 

on behalf of

Dr. Hans-Peter Simmen 

Academic Editor

PLOS ONE